# Analytical comparison of over-the-counter multiplex tests for influenza A, influenza B, and SARS-CoV-2

Sarah Lohsen,[1,2] Yana Hoy-Schulz,[2,3] Annie Taing,[1,2] Beth Coughlin,[1,2] Jordan Kim,[1,2] Heather B. Bowers,[4] Julie Sullivan,[3,4] Wilbur A. Lam,[3,4,5,6] Anuradha Rao,[3,4] Sarah W. Satola,[1,2,3] Gregory L. Damhorst[1,2,3]

**ABSTRACT**    Over-the-counter (OTC) lateral flow tests for respiratory viruses have newly emerged since the beginning of the coronavirus disease 2019 (COVID-19) pandemic and are increasingly available to consumers. While all marketed tests have met standards for Emergency Use Authorization (EUA), *de novo* classification, or 510(k) clearance, limited data are available to inform consumers of their relative performance. We performed a head-to-head benchtop analytical assessment of OTC tests available for purchase from retailers in the United States in the fall of 2025. Contrived specimen dilution panels were prepared from propagated viral stocks of influenza A H1N1pdm09, influenza A H3N2, influenza B (Victoria lineage), and severe acute respiratory syndrome coronavirus 2 (SARS-CoV-2) JN.1 (Omicron variant) in clinical matrix and applied to the swab provided with each kit. Tests were subsequently performed according to the manufacturer's instructions for use, and results were interpreted by two readers blinded to the starting test material. Lower limits at which three of three replicates of test material were detected were within a fourfold dilution for all four viruses and all eight OTC tests evaluated. We found that at low viral concentrations, many OTC tests were interpreted as negative at the start of the stated interpretation window but converted to a positive result by the end of the interpretation window. We conclude that eight OTC tests that are currently readily available to consumers perform similarly in a contrived specimen analytical study, but we recommend that users of OTC lateral flow tests allow for the full incubation time before concluding that a test is truly negative.

**IMPORTANCE** Our study provides consumers and medical professionals alike with a head-to-head comparison of the available over-the-counter coronavirus disease (COVID)/flu rapid antigen tests through a controlled, systematic evaluation. We also highlight the observation that, at low viral concentrations, many of the tests converted from negative to positive during the designated "interpretation window," resulting in a practical recommendation that consumers should wait for the full incubation time before concluding that a self-test is negative.

**KEYWORDS**    SARS-CoV-2, influenza, self-testing

S elf-tests for respiratory viral infections facilitate increased access to diagnosis, faster turnaround time, and lower costs despite lower sensitivity relative to laboratory-based molecular diagnostics (1). While self-tests, such as urine pregnancy tests, have existed for decades (2), over-the-counter (OTC) rapid tests for respiratory viruses did not become routine until the coronavirus disease 2019 (COVID-19) pandemic when they served an essential role in pandemic response (3). Shortly after the emergence of severe acute respiratory syndrome coronavirus 2 (SARS-CoV-2) self-tests, combination tests for influenza A, influenza B, and SARS-CoV-2 were developed. The first influenza/SARS-CoV-2 combination self-test marketed for OTC home use was authorized by the U.S. Food and

Address correspondence to Gregory L. Damhorst, gregory.damhorst@emory.edu.

Sarah Lohsen and Yana Hoy-Schulz contributed equally to this article. Author order was determined by seniority.

The authors declare no conflict of interest.

See the funding table on p. 8.

Drug Administration (FDA) under Emergency Use Authorization (EUA) in 2023 (4). Ten additional combination tests have since achieved EUA or traditional FDA clearance by the fall of 2025 (Table 1).

Of currently authorized or cleared tests in the United States, eight influenza/SARS-CoV-2 combination tests are readily available to consumers via online marketplaces and/or retail stores. All eight are lateral flow rapid antigen tests. Two additional tests, which are based on nucleic acid amplification technologies and authorized under EUA, have limited or no availability to consumers in the fall of 2025: the Lucira by Pfizer, production of which has been discontinued (9) and Aptitude Metrix, which is currently only available to healthcare providers despite its OTC authorization.

While some current self-tests were available during the 2024–2025 respiratory viral season, influenza/SARS-CoV-2 combination self-tests are still a nascent technology for consumers, and their real-world performance remains to be robustly described. Until such clinical data become available, we sought to compare analytical performance through head-to-head comparison of on-market influenza/SARS-CoV-2 self-tests. We used an abbreviated dilution series protocol derived from benchtop analytical studies employed in the evaluation and validation of *in vitro* diagnostics (10, 11). These "range-finding" exercises provide rapid insight into test performance and may aid consumers when understanding the range of assay performance and selecting OTC diagnostics. We selected viral strains reflecting recent or predicted subtypes or lineages of influenza A, influenza B, and SARS-CoV-2. Propagated stock viruses were diluted serially in panels in clinical matrix, which were then tested with eight OTC lateral flow assays interpreted by blinded test readers to determine the relative performance of marketed tests.

## MATERIALS AND METHODS

### Viral stocks

A summary of viral stocks utilized in this study and viral titers provided by manufacturers is provided in Table 2. Influenza A(H1N1)pdm09 strain A/Victoria/4897/2022, influenza A(H3N2) strain A/Croatia/10136RV/2023, and influenza B(Victoria lineage) B/Austria/1359417/2021 were provided by the Center for Biologics Evaluation and Research, Division of Biological Standards and Quality Control through a partnership with the Center for Devices and Radiological Health of the United States FDA and utilized for this study with permission. These strains reflect World Health Organization (WHO) recommendations for influenza vaccines in the 2025–2026 influenza season in the northern hemisphere. The following reagent was obtained through Biodefense and Emerging Infections (BEI) Resources, National Institute of Allergy and Infectious Diseases (NIAID), National Institutes of Health (NIH): SARS-Related Coronavirus 2, Isolate hCoV-19/USA/New York/PV96109/2023 (lineage JN.1; Omicron variant), NR-59693 lot 70064836.

### Rapid antigen tests

Seven of eight rapid antigen tests (Table 1) were purchased through Amazon.com. The iHealth COVID-19/Flu A&B Rapid Test was purchased from ihealthlabs.com. Manufacturer, regulatory status at the time of testing, and private label marketing in the United States corresponding to each device, as could be determined from the Devices@FDA database, are listed in Table 1.

### Dilution panel preparation

Nasal wash matrix (991-26-P) was purchased from Medix Biochemica USA, Inc. and confirmed to be negative for viral RNA with the Xpert Xpress CoV-2/Flu/RSV plus on the GeneXpert platform (Cepheid) and SeeGene Novaplex RV Master Assay prior to constructing dilution panels. Serial 1:10 major dilutions of viral stock and intermediate

**TABLE 1** Over-the-counter rapid antigen tests authorized or cleared by the US FDA by December 2025[a]

| Manufacturer | Labeling | Authorization or clearance | Read window | | Lot number(s) |
|---|---|---|---|---|---|
| | | | Start (min) | Stop (min) | |
| CorDx, Inc. | CorDx TyFast Flu A/B & COVID-19 At Home Multiplex Rapid Test | EUA | 10 | 30 | US2524803 |
| ACON Laboratories, Inc. | Flowflex Plus COVID-19 and Flu A/B Home Test[c] | 510 (k) | 15 | 30 | CFL4090003 |
| Healgen Scientific, LLC | Healgen Rapid Check COVID-19/Flu A&B Antigen Test* | De novo | 15 | 20 | 2505265DEN |
| | InBios COVID-19/Flu A&B RespiraDX Rapid Self-Test (InBios International, Inc.) | | | | |
| | INDICAID COVID-19/Influenza A&B Antigen Test (PHASE Diagnostics, Inc.) statID PRO COVID-19/Flu A&B (Meridian Bioscience, Inc.) | | | | |
| | Equate COVID-19 & Flu A/B Antigen Test (Walmart, Inc.) | | | | |
| | GenaCheck COVID-19/Flu A&B Rapid Self-Test (Genabio Diagnostics, Inc.) | | | | |
| | Consult COVID-19/Flu A&B Antigen Home Test (McKesson Medical-Surgical, Inc.) | | | | |
| | Rite Aid COVID + Flu A/B Antigen Test (RITE AID) | | | | |
| | ACCUBIO COVID-19/Flu A&B Antigen Test (Shanghai Douglas Medical Device, Co., Ltd.) healthconfirm COVID-19/Flu A&B Antigen Test (Confirm Biosciences) | | | | |
| | Rapid Response Influenza AB + COVID-19 Antigen Detection Test (BTNX, Inc.) | | | | |
| | RapidGo Flu A/B & Covid-19 Home Test (Kitlab, Inc.) | | | | |
| | Walgreens COVID-19 & Flu A +B Antigen Test (WALGREEN CO.) | | | | |
| | CVS Health Combo COVID-19 & Flu A/B Antigen Test (CVS Pharmacy Inc) | | | | |
| iHealth Labs, Inc. | iHealth COVID-19/Flu A&B Rapid Test | EUA[b] | 15 | 30 | 242CF10828, 24CF10927 |
| SEKISUI Diagnostics, LLC | OSOM Flu SARS-CoV-2 Combo Home Test | EUA | 10 | 30 | 241420A |
| OSANG LLC | QuickFinder COVID-19/Flu Antigen Self Test | 510 (k) | 15 | 30 | 01CCA160 |
| | BinaxNOW COVID-19/Flu A&B Combo Self Test* | | | | |
| Watmind USA | Speedy Swab Rapid COVID-19 + Flu A&B Antigen Self-Test | EUA | 15 | 20 | 24,100,200 |
| Wondfo USA Co., Ltd. | WELLlife COVID-19 / Influenza A&B Home Test*,[d] | 510 (k) | 10 | 20 | WV01411010 |
| | INDICAID COVID-19/Flu A&B Home Test (PHASE Diagnostics, Inc.) | | | | |
| | Hough COVID-19/ Flu A&B Home Test (Hough Diagnostics Pty Ltd) | | | | |
| | 2San 3 in 1 COVID-19 + Flu A&B Rapid Self-Test (2San LLC) | | | | |
| Access Bio, Inc. | CareSuperb COVID-19/Flu A&B Antigen Combo Home Test[e] | 510 (k) | 10 | 15 | Not tested |

[a]The tests manufactured by Osang LLC, Healgen Scientific, LLC, and Wondfo USA Co., Ltd (5–7) are marketed in the United States with additional or alternative branding under private label manufacturing arrangements listed here. In these cases, the labeling for the product purchased for this investigation is denoted with an asterisk (*). Lot numbers used for this study are included in the far right column.
[b]The iHealth COVID-19/Flu A&B Rapid Test and iHealth Flu A&B/COVID-19/RSV Rapid Test received 510(k) clearance on 12/12/2025. The products used in this study were marketed under EUA at the time of purchase.
[c]Four-plex Flowflex Plus RSV + Flu A/B + COVID Home Test received 510(k) clearance 22 October 2025 but was not available for this study (8).
[d]Two-plex WELLlife Flu A&B Home Test received 510(k) clearance on 20 August 2025, but the three-plex WELLlife test was chosen for this study.
[e]The CareSuperb COVID-19/Flu A&B Antigen Combo Home Test was not available from retailers at the time of this study.

1:2 dilutions of each major dilution were prepared in negative nasal wash matrix. Next, 180 µL aliquots of each of these dilutions were frozen at −80°C.

## Assay testing

Testing was conducted in triplicate for each device and viral stock dilution. The desired aliquot was thawed, and 50 µL was carefully dispensed from a micropipettor onto the tip of the kit's provided swab. The LFA was then performed by following the kit's instructions for use (IFU). A nasal wash-only preparation without virus was included for each device in each batch of testing as a negative control and to facilitate blinding of individuals interpreting results. Test interpretations were recorded at the beginning and end of the manufacturer's indicated read window. Readers did not know whether the device was tested with a viral dilution or negative control, nor did they know the identity of the virus(es) being tested during that session. In most cases, two readers were employed to interpret the device result.

**TABLE 2** Viral stocks used to generate dilution series in clinical matrix for head-to-head testing of combination OTC rapid antigen tests[a]

| Virus | Lineage/subtype | Strain | Titer of stock material | Genome equivalents/mL |
|---|---|---|---|---|
| Influenza A | H1N1pdm09 | A/Victoria/4897/2022 | $10^{9.25}$ EID$_{50}$/mL | $1.72 \times 10^8$ |
| Influenza A | H3N2 | A/Croatia/10136RV/2023 | $10^{7.7}$ EID$_{50}$/mL | $3.16 \times 10^8$ |
| Influenza B | Victoria | B/Austria/1,359,417/2021 | $10^{8.5}$ EID$_{50}$/mL | $5.07 \times 10^8$ |
| SARS-CoV-2 | JN.1 (Omicron variant) | hCoV-19/USA/New York/ PV96109/2023 | $2.2^6$ TCID$_{50}$/mL | $1.12 \times 10^8$ |

[a]Genome equivalents/mL were calculated from a 1:100 dilution of the stock material.

Each LFA provides an interpretation window in its IFU before which and after which the results of the device cannot be validly interpreted (Table 1). The final interpretation in our primary analysis was from the reader who interpreted the greatest number of positive replicates at the end of the designated interpretation window allowed by the self-test instructions for use. Interpretations were initially recorded at the beginning of the interpretation window, but after conversion from negative to positive within the interpretation window was observed, endpoint observations were also recorded and adopted as the primary measure.

The testing strategy began with all devices being tested with a 1:100 dilution of viral stock in clinical matrix. In the case where the final interpretation included at least one positive result, the next 1:2 dilution in the sequence was performed, and increasingly more dilute specimens were tested until the final interpretation included three negative replicates. When all three replicates were interpreted as negative by both readers, additional dilutions were not performed. In the case where a 1:100 dilution did not yield three of three positive results, a 1:80 dilution (prepared by serial 1:2 dilution of a 1:10 dilution of viral stock in clinical matrix) was performed.

## Digital droplet PCR

To quantify viral content, digital droplet PCR (ddPCR) was performed using an aliquot of the 1:100 dilution of viral stock in clinical matrix for each dilution series. 140 µL of the material was extracted via the KingFisher Apex using the MagMAX Viral/ Pathogen Nucleic Acid Isolation Kit, and then eluted in 60 µL of the MagMAX Viral/ Pathogen Elution Buffer (Thermo Fisher). The eluate was frozen temporarily at −20°C before performing the ddPCR using the QX600 AutoDG Droplet Digital PCR System as previously described (12). Viral genome quantities for the stock material and additional dilutions were calculated from the ddPCR result obtained from the 1:100 dilution (Table 2). Primer and probe sequences for all three viruses are summarized in Table 3 (12–14).

## RESULTS

### Triplicate range-finding experiments

The results of dilution panel testing are depicted in Fig. 1. Lowest concentrations where three of three replicates were detected were within a fourfold dilution for all eight devices with each virus tested: 1:100–1:400 for influenza A(H1N1)pdm09, 1:200–1:800 for influenza A(H3N2), 1:80–1:200 for influenza B, and 1:100–1:400 for SARS-CoV-2. No singular test appeared to perform consistently better than all other tests, and the lowest detected concentrations were achieved by different devices with each virus. For influenza A(H1N1)pdm09, the lowest detected concentration was the 1:400 dilution detected by the WELLlife and OSOM tests. For influenza A(H3N2), the lowest detected concentration was the 1:800 dilution detected by the Healgen and SpeedySwab tests. For influenza B, the lowest detected concentration was the 1:200 dilution detected by the OSOM test. For SARS-CoV-2, the lowest detected concentration was the 1:400 dilution detected by the FlowFlex test.

**TABLE 3** Primers used for digital droplet PCR[a]

| Virus | Name | Sequence (5′ to 3′) |
|---|---|---|
| Influenza A | InfA Forward | CAA GAC CAA TCY TGT CAC CTC TGA C |
| | InfA Reverse | GCA TTY TGG ACA AAV CGT CTA CG |
| | InfA Forward 2 | CAA GAC CAA TYC TGT CAC CTY TGA C |
| | InfA Reverse 2 | GCA TTT TGG ATA AAG CGT CTA CG |
| | InfA Probe | /FAM/TGC AGT CCT /ZEN/ CGC TCA CTG GGC ACG/3IABkFQ/ |
| Influenza B | InfB Forward | TCC TCA AYT CAC TCT TCG AGC G |
| | InfB Reverse | CGG TGC TCT TGA CCA AAT TGG |
| | InfB Probe | /HEX/CCA ATT CGA /ZEN/ GCA GCT GAA ACT GCG GTG/ 3IABkFQ/ |
| SARS-CoV-2 | 2019-nCoV N2-F | TTA CAA ACA TTG GCC GCA AA |
| | 2019-nCoV N2-R | GCG CGA CAT TCC GAA GAA |
| | 2019-nCoV N2-P | FAM-ACAATTTGC CCC CAG CGC TTC AG-BHQl |

[a]Primer sequences for influenza were obtained from the CDC (13). SARS-CoV-2 primers were obtained from reference 14.

## Result conversion during the interpretation window

A secondary and practical observation of this study was the increase in sensitivity of results at the end of the LFA interpretation window, especially for weak positives. Seven of the eight tests exhibited at least one instance where a reader did not see a line at the beginning of the interpretation window but did see a line at the end of the interpretation window. Result conversion was observed with all rapid tests with a 10-min or longer interpretation window as well as the Healgen test at the lowest consistently detected concentration and highlights the potential for rapid tests to convert from negative to positive during the interpretation window for weak positives. Five tests were completely negative for the initial read and only positive after the final read at the lowest

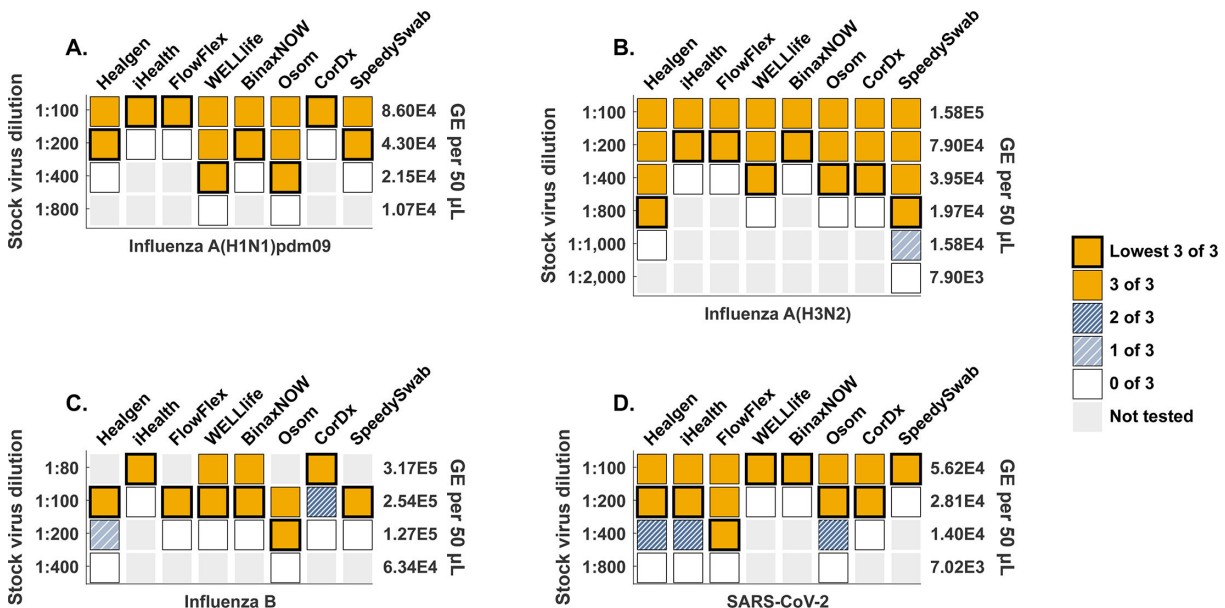

**FIG 1** Eight OTC influenza A/B and SARS-CoV-2 multiplex rapid tests exhibit similar analytical performance with a contrived dilution series. Test performance is summarized here for Healgen, iHealth, FlowFlex, WELLlife, BinaxNOW, CorDx, and SpeedySwab tests for (A) influenza A(H1N1)pdm09, (B) influenza A(H3N2), (C) influenza B, and (D) SARS-CoV-2. Gold squares indicate concentrations where 3 of 3 tests were read as positive. Dark blue squares indicate concentrations where two of three tests were read as positive. Light blue squares indicate concentrations where one of three tests was read as positive. White squares indicate concentrations where zero of three tests was read as positive. Light gray indicates concentrations for which tests were not performed. Bolded squares indicate the lowest concentration where three of three tests were read as positive. Concentration of viral genome equivalents (GE) is calculated from droplet digital (ddPCR) measurements of the 1:100 dilution and expressed here as GE per 50 µL, the volume used to spike each swab during LFA testing.

positive concentration with at least one virus, including iHealth for SARS-CoV-2, Flowflex for SARS-CoV-2, WELLlife for influenza A(H1N1)pdm09 and SARS-CoV-2, BinaxNOW for influenza A(H1N1)pdm09 and SARS-CoV-2, and OSOM for influenza A(H1N1)pdm09, influenza A(H3N2), and influenza B. These observations are presented in Fig. 2. These findings should be assessed in real-world settings with clinical samples.

## DISCUSSION

Although FDA marketing authorization or clearance is based on adequate performance in comparison to a predicate diagnostic test, relative test performance of OTC self-tests for influenza A, influenza B, and SARS-CoV-2 cannot be directly gleaned from data filed with the FDA. This is particularly important to consumers who lack highly visible and interpretable information about test performance when making purchasing decisions.

Furthermore, data submitted by test manufacturers are based on limited analyses of viral strains in the laboratory setting and clinical study data typically gathered during a single winter respiratory season in the United States. Viral evolution underlies the ability for common respiratory viruses to persist in the population, reinfect immunologically non-naïve human hosts, and make influenza A an ever-present pandemic threat. Antigen tests in particular may be negatively affected by even subtle viral evolution affecting the nucleoprotein antigen if changes alter the epitopes recognized by antibodies employed

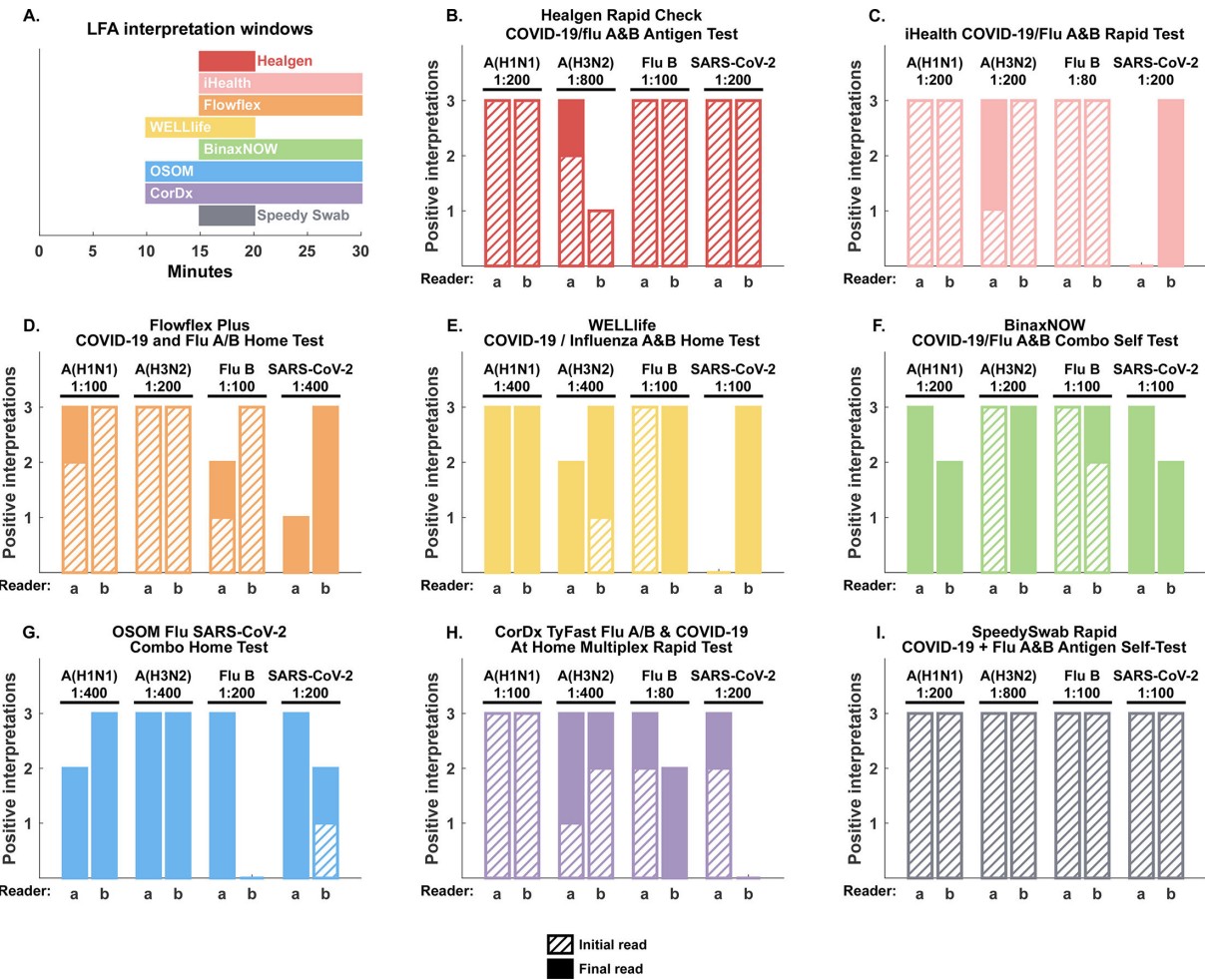

FIG 2 Result conversion was observed with seven of eight rapid tests at the lowest consistently detected concentration. (A) Graphical summary of assay interpretation windows described in package inserts. (B–I) Assay interpretation at the beginning (hashed bars) and end (solid face bars) of the interpretation window highlight the potential for rapid tests to convert from negative to positive during the interpretation window. Interpretations for both readers (A and B) are depicted for (B) Healgen, (C) iHealth, (D) FlowFlex, (E) WELLlife, (F) Binax, (G) OSOM, (H) CorDx, and (I) SpeedySwab.

in consumer lateral flow immunoassays, a phenomenon that was prominent during the H1N1 pandemic of 2009 (15, 16). For this reason, the FDA has advised test manufacturers to perform seasonal surveillance of marketed antigen-based diagnostic tests (17), and ongoing surveillance of test performance with emerging strains relevant to public health is needed.

Our NIH-supported Rapid Acceleration of Diagnostics (RADx) test validation center at Emory has previously examined the test performance of many of these rapid antigen tests with a strain of avian influenza A(H5N1) (10) isolated from a dairy cow in Ohio in 2024, a year that was notable for expanded spread of avian influenza among birds and mammals and spillover resulting in at least 77 confirmed or probable cases in humans (18). Here, we conducted a similar study to inform researchers, clinicians, and —most importantly—consumers regarding the relative performance of OTC rapid tests with seasonal influenza strains. This work was performed independent of NIH-sponsored programs and supported by Flu Lab (19).

Our data highlight overall similar analytical performance across all eight devices for four viral strains predicted to reflect those that will circulate during the 2025–2026 respiratory season. There are important caveats to our data, however. First, we performed only three replicates per dilution for each virus. While this testing approach mimics methods recommended by the Clinical Laboratory Standards Institute (CLSI) (20) and typically requested by the FDA, studies to determine the analytical limit of detection are typically carried out to 20 replicates with a requirement for 95% positivity rate at a given dilution (otherwise a more concentrated preparation must be tested). Therefore, our data are not as robust as standard analytical methods employed in diagnostic device regulatory and quality assessments. Despite this abbreviated approach, we still consider the present analysis to provide important insight into relative test performance that can inform test users.

An additional limitation of our approach is that we have tested only four viral strains. We chose influenza strains that are constituents of World Health Organization recommendations for 2024–2025 vaccines in the northern hemisphere. Meanwhile, the JN.1 isolate for SARS-CoV-2 was readily available through BEI resources and exhibits similarities to currently circulating SARS-CoV-2 variants. Despite these efforts to examine device performance with contemporary strains, relevance of these viruses can only be predicted at this time, and importantly, this testing did not include influenza A H3N2 subclade K, which dominated influenza circulation in the United States in late 2025 into early 2026 (21, 22). Future test efficacy surveillance programs at our institution are being developed to achieve "real time" evaluations using circulating strains.

A finding with potential clinical implications was observed in that the sensitivity of the tests was better at the end of the test interpretation window. All eight self-tests have interpretation windows specified by manufacturers during which results are considered valid and outside of which results are considered invalid. Interpretation window durations range from 5 to 20 min, beginning 10 or 15 min after the specimen is introduced and ending 20 or 30 min after the specimen is introduced (Table 1 and Fig. 2A). Results within these windows are stated to be valid in the provided IFU. However, we found that in most devices at lower viral concentrations near the limit of detections, tests were read as negative at the beginning of the interpretation window but had converted to positive by the end of the interpretation window. This finding did not impact the performance rating of the tests in our primary analysis since only results at the end of the interpretation window were considered. However, our observation of conversion from negative to positive during the interpretation window highlights the potential for false negative results if OTC tests are read only at the beginning of the interpretation window. It is unclear how much this in-window conversion phenomenon impacts result interpretation in real-world use, but as it has been generally well understood that LFAs can be poorly sensitive at the onset of symptoms, it is likely that many consumers are testing when actual nasal viral load is near the lower limit of detection. Thus, we highly

recommend users of OTC lateral flow tests should allow for the full incubation time before concluding they are indeed negative.

In conclusion, most lateral flow self-tests for influenza A, influenza B, and SARS-CoV-2 exhibit similar analytical performance in our abbreviated head-to-head dilution panel evaluation, and no singular device appeared superior to others in performance. The appearance of a line rendering a test result positive was observed during the stated interpretation window in seven of eight devices, suggesting that consumers should observe the test result at the end of the stated interpretation window to confirm a negative result. Ongoing surveillance of test performance will be required to ensure that on-market OTC self-tests continue to perform as expected with epidemiologically relevant viral strains and subtypes.

## ACKNOWLEDGMENTS

This work was performed independent of NIH-sponsored programs and was supported by Flu Lab. Through grants and investments, Flu Lab supports efforts to advance innovative approaches for the prevention and treatment of influenza.

Thanks to volunteer test interpreters: Juhi Patel, Andrea Torres-Ashline, Emilio Rodriguez, Soma Sannigrahi, Maria Camila Giraldo, Poonam Kanojiya, Erin Kowalsky, Sara Ping, Jayden Kimbro, Srija Vaidyanathan, Luyao Shen, Yih-Ling Tzeng, Ajay Balasubramaniam, Decoria Kimbrough, and Matthew Collins.

## AUTHOR AFFILIATIONS

[1]Investigational Clinical Microbiology Core, Department of Medicine, Emory University School of Medicine, Atlanta, Georgia, USA
[2]Division of Infectious Diseases, Department of Medicine, Emory University School of Medicine, Atlanta, Georgia, USA
[3]Center for the Advancement of Diagnostics for a Just Society (ADJUST Center), Emory University, Atlanta, Georgia, USA
[4]Department of Pediatrics, Emory University School of Medicine, Atlanta, Georgia, USA
[5]Wallace H. Coulter Department of Biomedical Engineering, Emory University School of Medicine and Georgia Institute of Technology, Atlanta, Georgia, USA
[6]Aflac Cancer and Blood Disorders Center of Children's Healthcare of Atlanta, Atlanta, Georgia, USA

## AUTHOR ORCIDs

Yana Hoy-Schulz  http://orcid.org/0009-0005-7904-4122
Anuradha Rao  http://orcid.org/0000-0001-6600-0677
Gregory L. Damhorst  http://orcid.org/0000-0002-2237-1713

## FUNDING

| Funder | Grant(s) | Author(s) |
| --- | --- | --- |
| Flu Lab | | Wilbur A. Lam |
| | | Gregory L. Damhorst |

## AUTHOR CONTRIBUTIONS

Sarah Lohsen, Data curation, Formal analysis, Investigation, Methodology | Yana Hoy-Schulz, Conceptualization, Formal analysis, Project administration, Validation | Annie Taing, Investigation | Beth Coughlin, Investigation | Jordan Kim, Investigation | Heather B. Bowers, Formal analysis, Investigation, Methodology | Julie Sullivan, Conceptualization, Project administration, Resources | Wilbur A. Lam, Conceptualization, Resources | Anuradha Rao, Methodology, Supervision | Sarah W. Satola, Investigation, Methodology,

Project administration, Supervision | Gregory L. Damhorst, Conceptualization, Formal analysis, Methodology, Supervision

## ADDITIONAL FILES

The following material is available online.

Open Peer Review

**PEER REVIEW HISTORY (review-history.pdf).** An accounting of the reviewer comments and feedback.

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
