## [Reviewer comments · Microbiology Spectrum]

Microbiology Spectrum

Analytical comparison of over-the-counter multiplex tests for influenza A, influenza B and SARS-CoV-2

Sarah Lohsen, Yana Hoy-Schulz, Annie Taing, Beth Coughlin, Jordan Kim, Heather Bowers, Julie Sullivan, Wilbur Lam, Anuradha Rao, Sarah Satola, and Gregory Damhorst

Corresponding Author(s): Gregory Damhorst, Emory University

Review Timeline:

Submission Date:	January 12, 2026
Editorial Decision:	February 25, 2026
Revision Received:	March 13, 2026
Accepted:	March 20, 2026

Editor: Karen Carroll

Reviewer(s): The reviewers have opted to remain anonymous.

Transaction Report:

DOI: <https://doi.org/10.1128/spectrum.00110-26>

Re: Spectrum00110-26 (Analytical comparison of over-the-counter multiplex tests for influenza A, influenza B and SARS-CoV-2)

Dear Dr. Gregory L Damhorst:

Thank you for the privilege of reviewing your work. Unfortunately, the second reviewer did not provide comments and therefore I am proceeding with the review of Reviewer #1 alone so as not to further delay the decision. I agree that the manuscript is well written but can be strengthened by addressing the reviewer's major comments.

Below you will find instructions from the Spectrum editorial office, and the reviewer's comments.

Revision Guidelines

Sincerely,
Karen Carroll
Editor
Microbiology Spectrum

Reviewer #1 (Comments for the Author):

Thank you for the opportunity to review the submission by Lohsen, et. al. In this manuscript, the authors perform an abbreviated analytical limit of detection study using eight direct-to-consumer home testing kits for influenza A/B and COVID. The authors

demonstrate relatively comparable performance of all assays across analytes. Perhaps the most valuable observation was the impact of confirming the test result at the end of the interpretation window as low concentration samples may have converted from negative to positive.

Overall the study is well written, the data is presented clearly, and the authors do not make any claims outside of the data they have shown.

Major Comment

The samples were quantified in EID₅₀ or TCID₅₀/ml. While this is acceptable for FDA submissions, product inserts, etc., providing the stock concentration in copies/mL might be a more suitable unit for clinical microbiologists to understand. I don't believe three replicates would be enough to perform probit calculations, but the authors could provide the quantitative level at which the LoD occurred when accounting for the dilution of the stock material. This could be performed by droplet digital PCR or running a standard curve with quantitative real-time PCR.

Minor Comment

Line 48: Add a reference for this sentence.

Dear Dr. Carroll,

Thank you for the opportunity to revise our manuscript and for the reviewer's comments. We have addressed these comments point-by-point below.

Reviewer #1 (Comments for the Author):

Thank you for the opportunity to review the submission by Lohsen, et. al. In this manuscript, the authors perform an abbreviated analytical limit of detection study using eight direct-to-consumer home testing kits for influenza A/B and COVID. The authors demonstrate relatively comparable performance of all assays across analytes. Perhaps the most valuable observation was the impact of confirming the test result at the end of the interpretation window as low concentration samples may have converted from negative to positive.

Overall the study is well written, the data is presented clearly, and the authors do not make any claims outside of the data they have shown.

Thank you, Reviewer #1, for reviewing our manuscript and for your helpful feedback!

Major Comment

The samples were quantified in EID₅₀ or TCID₅₀/ml. While this is acceptable for FDA submissions, product inserts, etc., providing the stock concentration in copies/mL might be a more suitable unit for clinical microbiologists to understand. I don't believe three replicates would be enough to perform probit calculations, but the authors could provide the quantitative level at which the LoD occurred when accounting for the dilution of the stock material. This could be performed by droplet digital PCR or running a standard curve with quantitative real-time PCR.

Thank you for this suggestion. We have added additional ddPCR measurements characterizing the 1:100 dilution, which was the starting point for LFA evaluation. Additional genome equivalents (GE)/mL are calculated from this measurement of the 1:100 dilution. Please note that we have also added an author as result of this additional work and added Table 3 with the primer/probe sequences used in these assays.

Minor Comment

Line 48: Add a reference for this sentence.

Thank you for pointing this out. We found a very relevant review published in 2024 in JCM and have cited that here.

Finally, we have also added slight updates to Table 1 after referencing the most recent updates to FDA databases.

Re: Spectrum00110-26R1 (Analytical comparison of over-the-counter multiplex tests for influenza A, influenza B and SARS-CoV-2)

Dear Dr. Gregory L Damhorst:

Thank you for responding to the reviewer's comments. Based upon my review of the revised manuscript, this manuscript can move forward to publication.

Your manuscript has been accepted, and I am forwarding it to the ASM production staff for publication. Your paper will first be checked to make sure all elements meet the technical requirements. ASM staff will contact you if anything needs to be revised before copyediting and production can begin. Otherwise, you will be notified when your proofs are ready to be viewed.

Sincerely,
Karen Carroll
Editor
Microbiology Spectrum